# Hypericin Ameliorates Depression-like Behaviors via Neurotrophin Signaling Pathway Mediating m6A Epitranscriptome Modification

**DOI:** 10.3390/molecules28093859

**Published:** 2023-05-03

**Authors:** Chunguang Lei, Ningning Li, Jianhua Chen, Qingzhong Wang

**Affiliations:** 1Institute of Chinese Materia Medica, Shanghai University of Traditional Chinese Medicine, Shanghai 201203, China; 2Shanghai Mental Health Center, Shanghai Jiao Tong University School of Medicine, Shanghai 200030, China

**Keywords:** N6-methyladenosine modification, hypericin, antidepressant, neurotrophin signaling pathway

## Abstract

Hypericin, one of the major antidepressant constituents of St. John’s wort, was shown to exert antidepressant effects by affecting cerebral CYP enzymes, serotonin homeostasis, and neuroinflammatory signaling pathways. However, its exact mechanisms are unknown. Previous clinical studies reported that the mRNA modification N6-methyladenosine (m6A) interferes with the neurobiological mechanism in depressed patients, and it was also found that the antidepressant efficacy of tricyclic antidepressants (TCAs) is related to m6A modifications. Therefore, we hypothesize that the antidepressant effect of hypericin may relate to the m6A modification of epitranscriptomic regulation. We constructed a UCMS mouse depression model and found that hypericin ameliorated depressive-like behavior in UCMS mice. Molecular pharmacology experiments showed that hypericin treatment upregulated the expression of m6A-modifying enzymes METTL3 and WTAP in the hippocampi of UCMS mice. Next, we performed MeRIP-seq and RNA-seq to study m6A modifications and changes in mRNA expression on a genome-wide scale. The genome-wide m6A assay and MeRIP-qPCR results revealed that the m6A modifications of Akt3, Ntrk2, Braf, and Kidins220 mRNA were significantly altered in the hippocampi of UCMS mice after stress stimulation and were reversed by hypericin treatment. Transcriptome assays and qPCR results showed that the Camk4 and Arhgdig genes might be related to the antidepressant efficacy of hypericin. Further gene enrichment results showed that the differential genes were mainly involved in neurotrophic factor signaling pathways. In conclusion, our results show that hypericin upregulates m6A methyltransferase METTL3 and WTAP in the hippocampi of UCMS mice and stabilizes m6A modifications to exert antidepressant effects via the neurotrophin signaling pathway. This suggests that METTL3 and WTAP-mediated changes in m6A modifications may be a potential mechanism for the pathogenesis of depression and the efficacy of antidepressants, and that the neurotrophin signaling pathway plays a key role in this process.

## 1. Introduction

Major depressive disorder (MDD) is one of the most severe mental illnesses, with prevalence rates ranging from 2.2% to 26.8% of the global population [1,2]. According to the World Health Organization (WHO), more than 800,000 people die by suicide each year due to depression [1,3]. However, the pathogenesis of depression remains unclear. In recent decades, the serotonin imbalance theory was widely accepted, and SSRIs were developed as antidepressants [4,5]. However, recent studies challenged the serotonin imbalance theory, indicating that not only is there no direct relationship between serotonin activity and depression, but depressed patients may develop a lifelong dependence on the drug based on the serotonin imbalance theory [6]. Therefore, it is necessary to explore new molecular mechanisms in the pathogenesis of depression and to develop new antidepressants. Epigenetic modifications of genes play an important role in the development of depression [7,8,9] and, similar to DNA methylation modifications and histone modifications, changes in the homeostasis of m6A modifications may also contribute to depression [10,11]. m6A modifications are among the most common modifications of RNA [12], and this dynamic and reversible modification primarily depends on regulation by methyltransferases and demethylases, including Methyltransferase-like protein 3 (METTL3), Methyltransferase-like protein 14 (METTL14) Wt1 Associated Protein (WTAP) Fat Mass and Obesity-Associated Protein (FTO) and Alkb Homolog 5 (ALKBH5) [11,13]. In eukaryotic organisms, m6A modifications are involved in biological processes, affecting RNA exit, shearing, translation, and degradation. Stable m6A modifications contribute to brain development [12,14]. Conversely, disruption of m6A modification may lead to the development of various diseases, including depression.

The occurrence of depression is closely related to the disruption of m6A modification of RNA caused by changes in the activity or expression of m6A-modifying enzymes [10,11,15,16]. Demethylase FTO expression was found to be markedly downregulated in the blood of depressed patients and in the hippocampus of mice with depressive-like behavior [15]. Additionally, targeted suppression of FTO expression in the hippocampus of mice showed depressive-like behaviors during tests, whereas over-expression of FTO mRNA provided antidepressant effects. Reduction in FTO activity directly increased m6A methylation modifications at Adrenoceptor beta 2 (Adrb2) mRNA, decreased Adrb2 mRNA stability, and altered the expression of Myc Proto-Oncogene (c-MYC) and Sirtuin 1(Sirt1) in downstream signaling pathways [15]. Chronic stress was reported to impair hippocampal synaptic plasticity in mice while decreasing protein levels of FTO [17]. Another methyltransferase, METTL3, was also associated with depression. The deletion of METTL3 in the hippocampus of mice can lead to decreased neurogenesis and depressive-like behaviors [18]. In addition, the antidepressant efficacy of tricyclic antidepressants (TCAs) was associated with m6A modifications [19]. Overall, these studies provided impressive evidence for a link between epitranscriptomic modifications, depressive-like behaviors, and antidepressant effects.

As an important natural antidepressant, St. John’s wort is widely used to treat mild to moderate depression because it has good clinical efficacy and fewer side effects than selective serotonin reuptake inhibitors (SSRIs) [20,21]. However, the mechanism of antidepressant action remains unclear. In addition, a recent study revealed that hyperforin, the active ingredient in St. John’s wort extract, inhibits the endothelial-to-mesenchymal transition by mediating the methyltransferase METTL3 [22]. We speculate that the antidepressant effect of hypericin may be related to m6A modification of epitranscriptomic regulation; however, no such studies were published. In the present study, we established an unpredictable chronic mild stress and orphan mouse model to investigate the antidepressant efficacy of hypericin. The expression of five classical methylation-modifying enzymes (METTL3, METTL14, WTAP, FTO, ALKBH5) was examined using molecular pharmacology assays, including MeRIP-seq, which can detect genome-wide m6A modification changes in the hippocampus by capturing RNA fragments with m6A antibodies, carrying out second-generation sequencing, and performing bioinformatic analysis to investigate the antidepressant mechanisms of hypericin in combination with the transcriptome data.

## 2. Results

### 2.1. Hypericin Relieve Depressive-like Behavior of UCMS Mice

To measure whether hypericin treatment can reduce the depressive-like behavior of UCMS mice, the TST, FST, OFT, and SPT were employed to measure the depressive-like behavior in the four groups (Control, UCMS, UCMS + duloxetine, UCMS + hypericin). The immobility time of both FST and TST was measured to determine the desperate behavior of tested mice. As shown in Figure 1A,B, the immobility time of the mice treated with hypericin significantly decreased compared with untreated UCMS groups (*p*-value < 0.01). The results of the OFT test showed that the total distance in the hypericin group significantly increased compared with UCMS, suggesting that hypericin can improve the autonomous mobility of mice with depressive-like behavior (Figure 1C, *p*-value < 0.001). The sucrose preference ratio, used as a measure of anhedonia behavior in the SPT test, and the sucrose preference ratio in the hypericin group significantly increased compared with those of UCMS mice (Figure 1D, *p*-value < 0.001). Hypericin could significantly improve depressive behavior in UCMS mice and is comparable to the effect of the positive drug duloxetine.

### 2.2. Hypericin Upregulated the Expression of METTL3 and WTAP

To explore the antidepressant efficacy of hypericin related to the m6A methylation, we examined mRNA and protein expression of these m6A enzymes in the hippocampus of the tested mice. Considering that METTL3, METTL14, and WTAP are core components of the methyltransferase complex, and FTO and ALKBH5 are major demethylases [11,13], it was reported that METTL3, FTO, and ALKBH5 were implicated in the pathology of depression [15,18,23]. Therefore, those five genes were selected as potential targets. Meanwhile, there were not consistent results of the changes in these potential genes (METTL3 and FTO) and more studies need to be validated. For METTL3, Xu et al. (2022) found that the METTL3-specific deletion in the mouse hippocampus can induce the phenotype of depression-like behaviors and spatial memory reduction [18]. In contrast, another study on METTL3 reported to significant upregulation in the hippocampus of UCMS rats and contributed to their cognitive deficits (2022) [24]. For the inconsistent findings of FTO, downregulation expression of FTO was found in the hippocampus of depressed patients and mice exhibiting depressive-like behaviors [15]. However, in another study, the FTO deficiency mice model may reduce anxiety and depressive-like behavior via changes in the gut microbiota [25]. Based on the results of qRT-PCR, the mRNA expression of METTL3, WTAP, and FTO genes altered in the comparison between UCMS mice and control. In addition, upregulated expression of METTL3, WTAP, and FTO was observed in the hypericin group (Figure 2A–E). We further examined the protein levels of these three genes in the hippocampus. The Western blotting results suggested that METTL3 and WTAP significantly decreased expression in the UCMS mice and hypericin can significantly reverse the reduction in METTL3 and WTAP proteins expression (Figure 2F). Interestingly, FTO showed no changes in any of the three groups. In the present study, we replicated the downregulation of METTL3 in the hippocampus tissue in UCMS mice, which was consistent with the finding that METTL3-specific deletion in the mouse hippocampus can induce depressive-like behaviors [18]. Interestingly, we also discovered that the expression of WTAP decreased significantly in the mice with depressive-like behavior, which were not shown in the previous studies. The above results suggest that the antidepressant effect of hypericin may be related to METTL3 as well as WTAP-mediated m6A methylation modifications.

### 2.3. Hypericin-Treated Stabled m6A Methylation Levels of Hippocampus

Using MeRIP-seq, we explored the genome-wide m6A modifications in the control, UCMS, and hypericin treatment. The differentially methylated peaks (DMPs) were identified by MeTPeak software (MeTPeak, San Antonio, TX, USA) from both UCMS vs. control and UCMS vs. hypericin treatment. Most DMPs were distributed in the 3′UTR, promoter (≤1 kb), and exons (Figure 3A). The enriched motifs in these groups all contained ‘GGAC’, which was consistent with the m6A feature motif (Figure 3B). The majority of m6A peaks were preferentially located in the 3′UTR and CDS region (Figure 3C). When comparing UCMS vs. control, DMPs associated with the depressive behavior type revealed that m6A modification of 1013 peak regions in transcripts increased and 1598 peaks regions in transcripts decreased after UCMS treatment (Figure 3D). Among the most highly ranking significant DMP, these DMPs were located in genomics regions, including Ncor2 (chr5:125026883-125030017, *p*-value = 1.995262 × 10^−11^, fold change = 0.45), Shank1 (chr7:44344989-44353269, *p*-value = 3.981072 × 10^−11^, fold change = 0.60), Muc6 (chr7:141636300-141641384, *p*-value = 5.011872 × 10^−11^, fold change = 4.46), and Adcyap1r1 (chr6:55497425-55498669, *p*-value = 9.332543 × 10^−9^, fold change = 0.54) (Appendix A), which presented the changes in peak numbers by IGV software (Itegrative Genomics Viewer, Cambridge, MA, USA).

For the comparison between UCMS and hypericin treatment used to stabilize m6A methylation profiles, there were 772 DMPs regions in increased transcripts and 713 DMPs regions in decreased transcripts (Figure 3E). Among the ranking top significant DMP, these DMPs were located with genomics regions, including Hspa9 (chr18:34938361-34940234, *p*-value = 2.344229 × 10^−7^, fold change = 0.54), Rab11fip5 (chr6:85347945-85348246, *p*-value = 7.943282 × 10^−7^, fold change = 0.29), and Zbtb7a (chr10:81149075-81149376, *p*-value = 2.089296 × 10^−6^, fold change = 0.59) exhibited via IGV, which was associated with antidepressants (Appendix A).

We conducted the overlapping analysis and identified 298 DMPs reflected in the hypomethylated status in mice displaying depressive-like behavior, and contrasting hypermethylation in the control and treated mice (Figure 4A). The annotated function of these DMPs from GO analysis enriched the biological functions, including glutamatergic synapse, postsynaptic density, and neuronal cell body (Figure 4B,C). The annotated function of these DMPs from KEGG analysis revealed that most genes were enriched in long-term potentiation, glutamatergic synapse, GABAergic synapse, as well as the cAMP signaling pathway and neurotrophin signaling pathway. The above results indicate that the homeostasis of m6A modification in the hippocampus of mice is disturbed after UCMS stimulation, and treatment with hypericin modulates the disturbed homeostasis of m6A modification.

### 2.4. Hypericin-Treated Regulated mRNA Expression of Hippocampus in UCMS Mice 

RNA-seq analysis was also employed to explore the mechanism of action of hypericin on depression from transcriptomic level. A total of 850 differentially expressed genes and 765 genes were identified through comparison between UCMS vs. control and UCMS vs. hypericin treatment, respectively. For the top-ranking genes of comparison between UCMS vs. control, Shank1, Syt7, and Igfbp5 showed the most significant changes, as previously published (Appendix A). After treatment with hypericin, 99 genes, including Ndufs6, Mef2d, Nr2f2, Rimbp2, Gpx3, and Pcdhga11, were reversed (Appendix A, Figure 5A) and annotated with KEGG pathway and GO analysis. Interestingly, these genes were enriched into the neurotrophin signaling pathway shared by the enriched pathways for the DMPs in the MeRIP-seq. (Figure 5B,C). The above results suggest that hypericin treatment can modulate abnormal mRNA expression in the hippocampus of UCMS mice.

### 2.5. Validation of Results Candidate Genes and DMPs 

We further examined the overlapping biological terms in the gene enrichment analysis of differentially methylated genes in the MeRIP-seq and differentially expressed genes in the mRNA-seq. The four most important pathways, including the apelin signaling pathway, aldosterone synthesis and secretion, RNA degradation, and neurotrophin signaling pathway, were found to be shared with the two enrichment analyses. Of these, the neurotrophin signaling pathway is a key pathway that influences the antidepressant efficacy of various drugs. We further analyzed the m6A methylation located in Akt3, Ntrk2, Braf, Map2k1, and Kidins220, which were enriched into the neurotrophin signaling pathway (Table 1, Figure 6). We also validated the differential expression of several genes (Sort1, Camk4, and Arhgdig) that enriched the neurotrophin signaling pathway based on the analysis of mRNA–seq (Table 2). The results of qPCR demonstrated that the expression of Camk4 and Arhgdig was downregulated in the UCMS mice; on the other hand, expression was upregulated in the hypericin-treated mice. Meanwhile, the expression Sort1 mRNA was upregulated in the UCMS mice, which validated the results of RNA-seq (Figure 5D). For MeRIP-qPCR, the results of validation of the m6A methylation level of neurotrophin signaling genes demonstrated that the m6A methylation level of Akt3, Ntrk2, Braf, and Kidins220 increased in the hypericin-treated mice, in accordance with the results of MeRIP-seq (Figure 4D). Additionally, Ntrk2 mRNA expression was also found to be reduced in UCMS and reversed after hypericin treatment, while Map2k1 and Kidins220 increased following hypericin treatment (Figure 4E). This suggests that the neurotrophin signaling pathway played a key role in the antidepressant efficacy of hypericin.

## 3. Discussion

In our study, we investigated the antidepressant efficacy of hypericin by establishing a UCMS mouse model in combination with behavioral tests, using duloxetine as a positive control for the antidepressant evaluation of hypericin and focusing on its antidepressant mechanism. The results of TST and FST showed that treatment with hypericin and duloxetine significantly reduced the time to giving up struggle in a despairing environment in UCMS mice, and the results of SPT demonstrated that treatment with hypericin and duloxetine restored the pleasure deficit in UCMS mice. In addition, SPT also reflected hypothalamic satiety mechanisms [26], suggesting that the occurrence of depression and the efficacy of antidepressants may be related to obesity. Studies based on clinical data found a negative correlation between depressive symptoms and body weight in humans [27], and SSRIs treatment was also associated with weight gain in depressed patients [28,29]. Interestingly, obesity was also found to be regulated by the FTO gene. Recent genetic studies revealed that four polymorphic variants in the FTO gene were associated with body mass index in schizophrenia patients [30]. Meanwhile, FTO, an important m6A demethylase was investigated, and it was determined that knockout FTO mice exhibited depressive-like behavior and that FTO expression may correlate with the antidepressant efficacy of fluoxetine [15]. These studies suggested the therapeutic effect of hypericin may be associated with m6A methylation modifications and obesity. In the present study, an OFT test was performed as a complement-behavior test to measure depressive behavior in the UCMS model, despite the fact that OFT was commonly used to detect anxiety-like behavior [31]. From the results from OFT, the spontaneous activity exploration behavior of UCMS mice increased significantly after treatment with hypericin and duloxetine. In conclusion, hypericin has comparable antidepressant efficacy to duloxetine.

The expression of multiple m6A enzymes were measured in the hippocampus of mice. It was noted that the hippocampus connected to medial habenula (MHb) via the hypothalamus and septal nuclei [32]. This connection of hippocampal complex and the lateral habenula (LHb) may involve into the projections of GABAergic from the lateral septum [33,34,35]. We found that mRNA and protein levels of the methyltransferase METTL3, WTAP, decreased in the hippocampus after UCMS stimulation in mice, and hypericin administration significantly increased the expression of both. METTL3, an important component of methyltransferases, is involved in several biological processes in the brain, including memory formation and consolidation, synaptic plasticity, and neural stem cell differentiation [36,37]. Recently, Mareen Engel et al. (2018) found the gene expression of METTL3 was downregulated in the medial prefrontal cortex and the central amygdala of the stressed mice [36]. METTL3-deficient mice exhibited reduced hippocampal neurogenesis, which contributes to depressive behavior [18]. These results suggested that the antidepressant effect of hypericin may be related to altered expression of METTL3 and WTAP.

After discovering that hypericin increased the expression of methyltransferase METTL3 and WTAP in the hippocampus of UCMS mice, we examined the changes in m6A modification and mRNA expression in mouse hippocampal tissue on a genome-wide scale. The results of MeRIP-seq and RNA-seq showed that after UCMS stimulation, the homeostasis of m6A modification in mouse hippocampus was disturbed and the expression levels of some mRNAs changed. After hypericin treatment, the homeostasis of m6A modification was regulated and mRNA expression levels were modulated. Subsequent biological functional enrichment analysis showed that the various mRNAs and the DEGs were enriched in the apelin signaling pathway, aldosterone synthesis and secretion, RNA degradation, and neurotrophin signaling pathway. The neurotrophin signaling pathway included four types of neurotrophins, including nerve growth factor (NGF), brain-derived neurotrophic factor (BDNF), neurotrophin-3 (NT-3), and neurotrophin-4 (NT-4) [38,39]. Its biological function was related to regulate neuronal development, including synapse formation and synaptic plasticity [40,41]. Increasing evidence suggested that it mediated the onset of depression and the action of many antidepressants [42,43,44].

DEGs (Akt3, Ntrk2, Braf, Map2k1, and Kidins220) and differential genes (Sort1, Camk4, and Arhgdig) were extracted from the neurotrophin signaling pathway. In conjunction with subsequent validation, we found that in mice stimulated with UCMS, m6A modification of Akt3, Ntrk2, Braf, and Kidins220 mRNAs decreased after UCMS stimulation and increased after hypericin treatment. With respect to Ntrk2, we confirmed that m6A methylation and mRNA expression of Ntrk2 were reduced in the hippocampus of UCMS mice and that expression was reversed after hypericin treatment. As a receptor for BDNF, Ntrk2 is key to the efficacy of many antidepressants [45,46]. It was suggested that combined treatment with fluoxetine and melatonin may have a synergistic antidepressant effect, restoring BDNF-Ntrk2 signaling in the hippocampus [47]. In addition, the antidepressant efficacy of (R)-ketamine was also strongly associated with Ntrk2 [48]. Meanwhile, as the center of reward and aversion, the habenula tissue was implicated into the pathophysiology of major depression and antidepressant effects [49]. In rat and mouse models of depression, it was reported that antidepressant actions of ketamine mainly mediate the blockade of bursting activity in the lateral habenula neurons and relieve behavioral despair and anhedonia [50]. Additionally, Carolin Hoyer et al. (2012) found that deep brain stimulation treatment of the lateral habenula increased BDNF serum levels and exhibited an antidepressant effect [51]. This suggested that NTRK2 and BDNF may play a key role in habenula and could be associated with the antidepressant efficacy of ketamine. In conclusion, the dysregulated expression of Ntrk2 played a contributory role in depression and the targeting of novel antidepressants. 

In addition to genes enriched in the neurotrophic factor signaling pathway, we also focused on a number of other genes with large differential variation. We found that dysregulated expression of the mRNAs of Shank1, Syt7, and Igfbp5 could be associated with depressive behavioral phenotypes. Interestingly, the methylation modification of Shank1 was also significantly altered in the UCMS mice. It was reported that Shank1, as a gene associated with neuroplasticity, was decreased in the medial prefrontal cortex and hippocampus of mice with depressive behavior [52,53]. Although these changes in gene expression could not be reversed by hypericin, they are closely related to the onset of depression and could be potential targets for antidepressants.

Apart from the m6A modification mechanisms related to hypericin antidepressant effect, it was also found that hypericin can potently inhibit the activities of P450 enzymes (CYP) in in vitro experiments [54] and modify the levels of dopamine and serotonin in synaptic clefts [55,56]. Recent studies also showed that hypericin can exert antidepressant effects by inhibiting neuroinflammation. Zhai et al. (2022) found that hypericin can effectively alleviate the symptoms of postpartum depression in rats by inhibiting NLRP3 inflammasome activation and regulating glucocorticoid metabolism [57]. Another study also reported that chlorogenic acid and hypericin can exert antidepressant effects via the gut microbiota–neuroinflammatory axis [58]. Additionally, neuroinflammation can sustain the activation of the brain immune cell microglia and recruit other immune cells into the brain, a process which is typically associated with depression [59,60]. Neuroinflammation and immunological responses can be modulated by the STING gene and its pathways in the central nervous system. Administration of STING agonist can ameliorate stress-driven depression-like behaviors through the activation of microglial phagocytosis and suppression of neuroinflammatory cytokines [61]. The use of immunomodulatory ibrutinib can alleviate neuroinflammation and synaptic defects and have antidepressant effect in the LPS-induced depressive-like behavior mice [59].These above studies suggested that hypericin may produce antidepressant effects through multiple potential mechanisms. However, it is unclear whether the expression of key genes and molecular targets in those hypericin-related mechanisms are regulated by m6A modifications; this warrants further investigation. 

## 4. Conclusions

In conclusion, our study found that hypericin improved depression-like behaviors in UCMS mice, while upregulating METTL3 and WTAP expression in the hippocampi of UCMS mice and modifying the neurotrophin signaling pathway through m6A modification to exhibit antidepressant effects. The present study suggested that the m6A-modifying enzyme METTL3, as well as WTAP, may play a key role in the pathogenesis and treatment of depression, which provided some hints for the development of new antidepressant drugs at the epitranscriptomic level. Another important finding is that the antidepressant efficacy of hypericin is closely related to the neurotrophic factor signaling pathway, suggesting that the neurotrophic factor signaling pathway may be a potential mechanism for antidepressants.

## 5. Materials and Methods

All the experimental details can be found in the Appendix A.

### 5.1. Chronic Unpredictable Mild Stress Model and Treatments

Firstly, 4-week-old male C57BL/6 mice were bred for 2 weeks and used for subsequent experiments. All animal experiments were performed in accordance with a protocol approved by the University Animal Care and Use Committee at SHUTCM (PZSHUTCM210608001) and in accordance with the National Institutes of Health Guide for the Care and Use of Laboratory Animals. All mice were housed on a 12 h light–dark cycle at 25 ± 1 °C and received standard chow and water ad libitum.

The unpredictable chronic mild stress (UCMS) model was used in mice following a previously described protocol with minor modifications [62]. Briefly, mice in the control group were housed five to a cage, whereas mice in the UCMS group were housed individually. The mice in the UCMS group were randomly exposed to 9 different stresses for 21 days. These stresses primarily included tail pinching (2 min), food deprivation (24 h), overnight bed deprivation, water deprivation (12 h), wet bedding (12 h), tilting the cage 45° (24 h), nighttime lighting (12 h), noise (20 min), and body restraint (2 h). Mice in the control group were housed under normal conditions. After the three-week treatment, the mice in the UCMS group were randomly divided into three groups (*n* = 8 in each group): UCMS group, UCMS + duloxetine (duloxetine, 30 mg/kg, oral administration) group, and UCMS + hypericin (hypericin, 2.4 mg/kg, oral administration) group; in addition, the mice in the control group were treated with a similar amount of saline.

### 5.2. Behavioral Tests

#### 5.2.1. Sucrose Preference Test (SPT) 

The mice were allowed to drink from two bottles for two days in advance. On the first day, the mice were given two bottles of 1% sucrose solution, and on the second day, one bottle was filled with water and the other with 1% sucrose solution. The mice were not given water or food for 24 h before SPT. Then, the SPT with two bottles (one filled with water and the other with 1% sucrose solution) was given to the mice to drink for 24 h, and the position of the two bottles was exchanged after 12 h. The ratio of sucrose solution consumed was calculated as sucrose preference using the formula (sucrose intake/(sucrose intake + water intake) × 100).

#### 5.2.2. Forced Swimming Test (FST) 

Mice were placed individually in a glass bucket (height 30 cm, diameter 20 cm) filled to a height of approximately 20 cm with water 25 ± 1 °C. During the test, the mice were allowed to swim freely. The time of immobility was determined from video recordings of the last 4 min of the 6 min test period. 

#### 5.2.3. Tail Suspension Test (TST) 

In TST, mice were suspended on a horizontal bar for the test duration of 6 min. Next, the immobility time was estimated for the last 4 min with no active movements.

#### 5.2.4. Open Field Test (OFT) 

For spontaneous activity exploration during the OFT test, the total distance of autonomous movement in the box and time spent in the middle area were recorded using the EthoVision video tracking system (Noldus Information Technology™, Leesburg, VA, USA). Each mouse was placed individually in the center of an open field (50 × 50 × 50 cm) and observed in the open field for 5 min.

### 5.3. Tissue Isolation and mRNA Sequencing 

Total RNA was extracted from the hippocampus using TRIzol (Thermo Fisher Scientific, Waltham, MA, USA) and measured using OD260/OD280. RNA samples were stored at −80 °C for the subsequent experiments. 

Total mRNA was extracted with oligo (dT) from 2 ug total RNA. RNA libraries were then prepared using VAHTS Universal V8 RNA-seq Library Prep kits according to the manufacturer’s instructions. The quality of the libraries was reviewed and quantified using the BioAnalyzer 2100 system (Agilent Technologies, Santa Clara, CA, USA) and sequenced using the Illumina Novaseq platform. For mRNA sequencing data analysis, fastqc (version 0.19.11) was used to clean the raw data, and hisat2 was used to map the cleaned reads to the reference genome. Then, featureCounts was used to count the reads mapped to each gene, and the differential mRNA level was analyzed using the DESeq2 R package.

### 5.4. MeRIP Sequencing 

#### 5.4.1. Library Construction

The m6A RNA Enrichment Kit (Epigentek, Farmingdale, IL, USA) was used to perform MeRIP assays according to the manufacturer’s protocols. We incubated 10 ug of total mRNA with immunocapture buffer containing Immuno Capture Buffer, 2 ug of m6A antibody (Synaptic Systems), and affinity protein G beads for 90 min at room temperature under rotation so that the RNA with m6A modification binds to the affinity protein G beads and forms immunoprecipitation complexes, followed by the cleaved step with nuclear digestion enhancer and cleavage enzyme mix. The immunoprecipitation complex was then washed three times with buffer and digested with proteinase K for 15 min at 55 °C to release the enriched RNA from the magnetic beads. To purify and recover the enriched RNA, RNA binding particle analysis was performed. After the m6A-containing fragment RNA was enriched, the m6A-seq library was constructed using the SMARTer Stranded Total RNA-Seq Kit-v2 (634413, Takara, Japan). Additionally, the library preparations were sequenced on an Illumina Novaseq platform with a paired-end read length of 150 bp according to the standard protocols.

#### 5.4.2. Data Analysis 

Fastqc software was used to remove raw reads containing adapters, ploy-N, and inferior reads. The STAR aligner was used to align high-quality clean reads with the mouse genome (mm10) in subsequent analysis. Exomepeak2 was used to determine the m6A peaks in each anti-m6A immunoprecipitation group using the corresponding input samples as internal controls. Low-quality peaks were removed (peaks contained in 70% of the sample were retained). The distribution of m6A peaks was visualized using the Guitar package. ChIPseeker was used to examine the peaks in the genomic functional regions, including the promoter, 5’ UTR, 3’ UTR, exon and intron regions, and the functions of the peaks were annotated with TxDb.Mmusculus.UCSC.mm10.knownGene. Finally, the m6A-enriched motifs of each group were analyzed using HOMER.

#### 5.4.3. Differential Peaks Identification and Functional Analysis

Differential peak analysis was performed using MeTPeak [63] and the threshold of significance was set as *p*-value < 0.05 and FC > 1.5 (FC < 0.67). For the differential peaks and the located genes, biological function was annotated using Gene Ontology (GO) and Kyoto Encyclopedia of Genes and Genomes (KEGG) enrichment analysis.

### 5.5. qRT-PCR and MeRIP-qPCR

cDNA was synthesized from the 1ug mRNA with HiScript II Reverse Transcriptase (Vazyme, R232). Real-time PCR was performed with Universal SYBR qPCR Master Mix (Vazyme, Q511). β-ACTIN gene was used for the internal control and the 2^^(−ΔΔCt)^ methods were used to calculate the relative level of each mRNA. The sequence of PCR primers is shown in Appendix A.

The m6A RNA Enrichment Kit (Epigentek, Farmingdale, IL, USA) was applied to perform the MeRIP assays in line with the manufacturer’s protocols. The 10ug total RNA were input and 2 ug m6A antibody were immunoprecipitated in the MeRIP-qPCR, which is consistent with MeRIP-seq. HiScript II Reverse Transcriptase (Vazyme, R232) and AceQ^®^ Universal SYBR qPCR Master Mix (Vazyme, Q511) were, respectively, applied to perform reverse transcription and real-time qPCR. The primers of MeRIP-qPCR were designed according to the identified m6A peak in differential peak analysis. The mRNA of the corresponding sample without m6A immunoprecipitation was used as the internal control. Additionally, the 2^^(−ΔΔCt)^ methods were used to calculate the relative level of each mRNA. Primer sequences are shown in Appendix A.

### 5.6. Western Blot Analysis

Total hippocampal protein was extracted using RIPA lysis buffer and quantified using a protein analysis assay kit. Subsequently, a total of 20 ug protein of each sample was separated using 10% and 12% SDS-PAGE gels via electrophoresis. The proteins were then transferred to a polyvinylidene fluoride (PVDF) membrane, blocked with a buffer containing 5% nonfat milk for 2 h, and incubated overnight at 4 °C with primary antibodies. The primary antibodies were listed as follows: anti-METTL3 (1:2000, Pro-teintech, 15073-1-AP), anti-FTO (1:2000, Proteintech, 27226-1-AP), anti-WTAP (1:5000, Proteintech, 60188-1-Ig), β-ACTIN (1:50,000, Proteintech, 81115-1-RR). Subsequently, the proteins were hybridized with the secondary antibodies for 1 h and protein binding was detected using chemiluminescence methods.

### 5.7. Statistical Analysis

In this study, the number of animals selected for behavioral testing was determined based on our previous experimental experience (eight animals per group), and qPCR, MeRIP-qPCR, and Western blot were repeated at least three times independently. All data were calculated with the mean value and standard deviation. First, we used the Shapiro–Wilk normality test to determine whether the individual data were normally distributed. Then, Bartlett’s test was performed to ensure that the individual data had the same SDs. If the individual data followed a normal distribution and the group variances were homogeneous, we used the one-way ANOVA with Dunnett’s post hoc test. When group variances were not homogeneous, we used the Brown–Forsythe and Welch ANOVA test with Dunnett’s post hoc test. All statistical analyses were performed using GraphPad Prism 9 software (GraphPad, San Diego, CA, USA). For histological data analysis, we utilized Fastqc, Hisat2, DESeq2, STAR aligner, ChIPseeker, Exomepeak2, and MeTPeak for data cleaning, quality control, genomic matching, and differential analysis. A threshold of *p* < 0.05 was considered statistical significance.

## Figures and Tables

**Figure 1 molecules-28-03859-f001:**
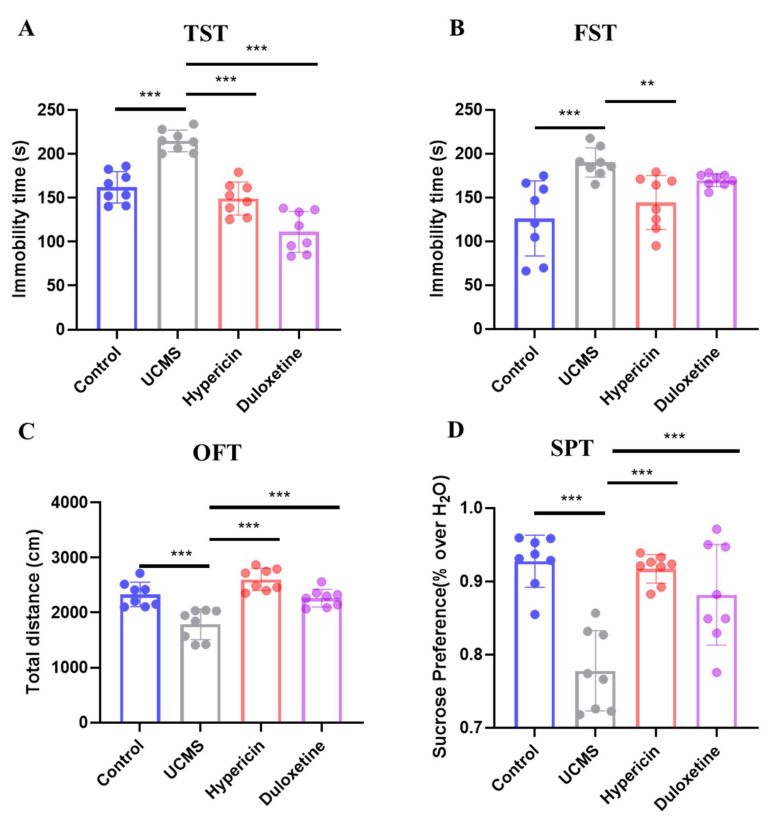
Hypericin treatment reduced the depressive-like behaviors of UCMS mice. (**A**,**B**) Time of immobility of mice in the TST and FST separately. (**C**) The total distance of mice in the OFT. (**D**) Sucrose preference of mice in the SPT. Data are expressed as mean ± S.D. N = 8 for each group. **, *p* < 0.01; ***, *p* < 0.001 vs. Control group. Blue represents the control group, gray represents the UCMS group, red represents the hypericin group and purple represents the duloxetine group.

**Figure 2 molecules-28-03859-f002:**
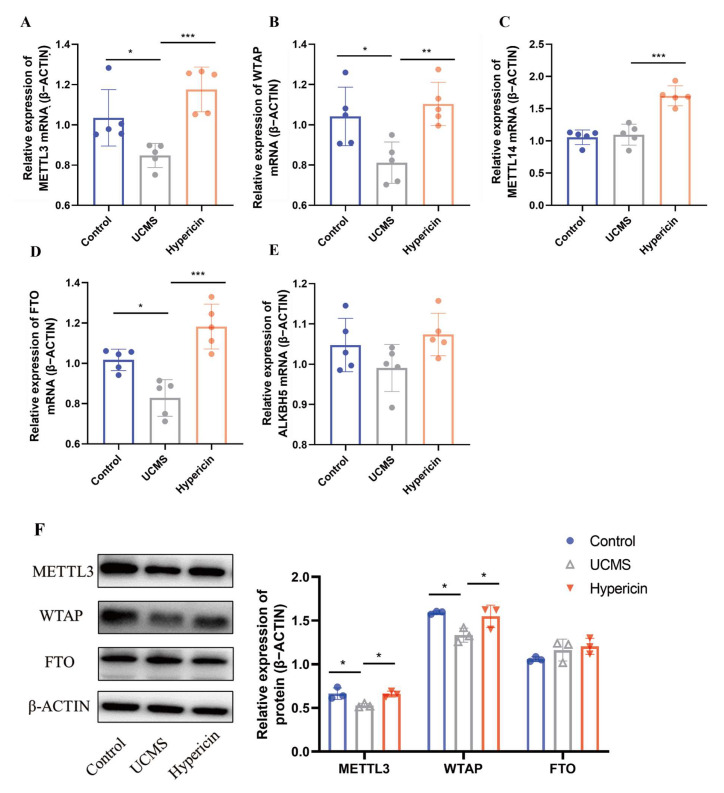
(**A**–**E**) The mRNA expression of METTL3, FTO, WTAP, METTL14, and ALKBH5 in the hippocampus of control, UCMS, and hypericin-treated mice (*n* = 5 mice per group). (**F**) METTL3, WTAP, and FTO protein expression in the hippocampus of control, UCMS, and hypericin-treated mice (*n* = 3 mice per group). Data are expressed as mean ± S.D. *, *p* < 0.05 **, *p* < 0.01; ***, *p* < 0.001 vs. Control group. Blue represents the control group, gray represents the UCMS group, and red represents the hypericin group.

**Figure 3 molecules-28-03859-f003:**
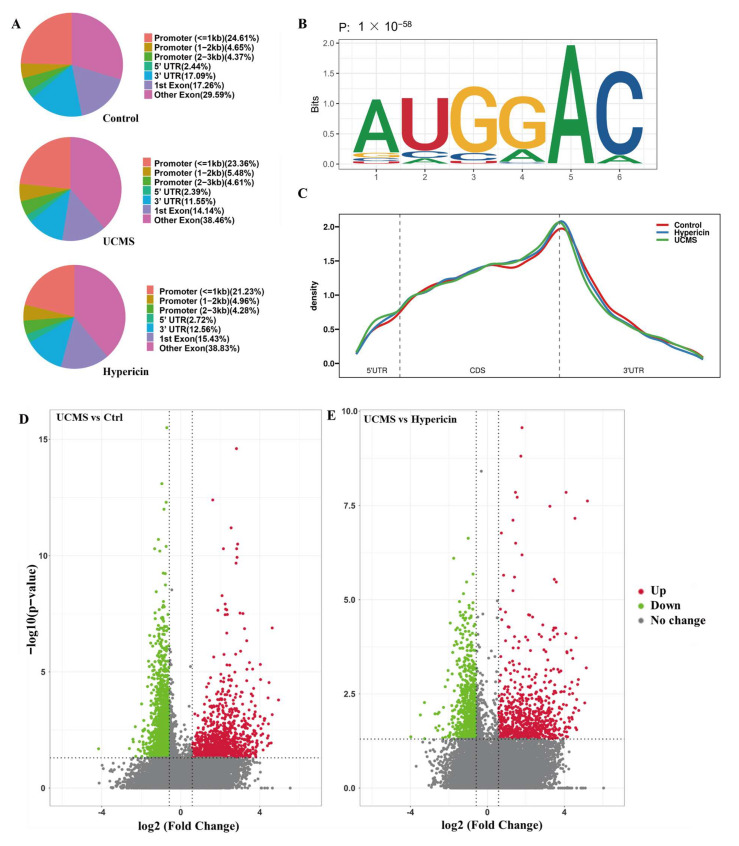
(**A**) Annotation of peaks detected in in the hippocampus of control, UCMS, and hypericin-treated mice. (**B**) The most abundant motifs detected in peaks. (**C**) Distribution of m6A peaks throughout the whole mRNA transcript among the groups. (**D**) Differentially methylated peaks among control and UCMS groups (*n* = 3 of control, *n* = 3 of UCMS). (**E**) Differentially methylated peaks among hypericin and UCMS groups (*n* = 3 of UCMS, *n* = 2 of hypericin).

**Figure 4 molecules-28-03859-f004:**
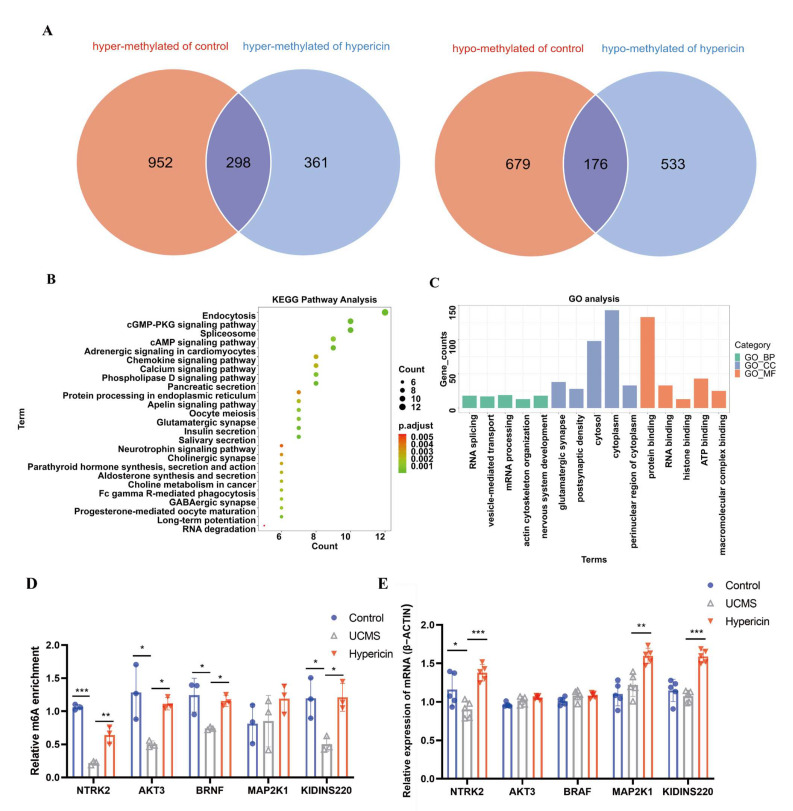
(**A**) Left: Overlap of the genes that are hypermethylated in control and hypericin-treated mice and hypomethylated in UCMS mice. Right: Overlap of the genes that are hypomethylated in control and hypericin-treated mice and hypermethylated in UCMS mice. (**B**,**C**) KEGG pathway analysis and GO analysis of hypermethylated genes in control and hypericin-treated mice and hypomethylated in UCMS mice. (**D**) Validation of m6A methylation level of genes enriched in the neurotrophin signaling pathway in the hippocampus among three groups (*n* = 3 mice per group). (**E**) mRNA level of genes enriched in the neurotrophin signaling pathway in the hippocampus among there groups (*n* = 5 mice per group). Data are expressed as mean ± S.D. *, *p* < 0.05 **, *p* < 0.01; ***, *p* < 0.001 vs. Control group.

**Figure 5 molecules-28-03859-f005:**
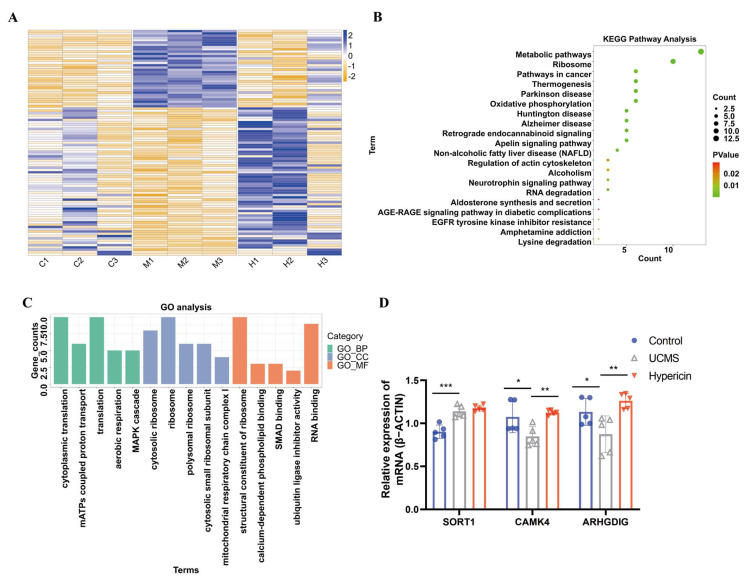
(**A**) Heatmap of differentially expressed genes among the groups. (**B**,**C**) KEGG pathway analysis and GO analysis of hypermethylated genes among control and hypericin groups. Abbreviation: mitochondrial ATP synthesis (mATPs). (**D**) Validation of mRNA level of genes enriched in the neurotrophin signaling pathway in the hippocampus among the groups (*n* = 5 mice per group). Data are expressed as mean ± S.D. *, *p* < 0.05 **, *p* < 0.01; ***, *p* < 0.001 vs. Control group.

**Figure 6 molecules-28-03859-f006:**
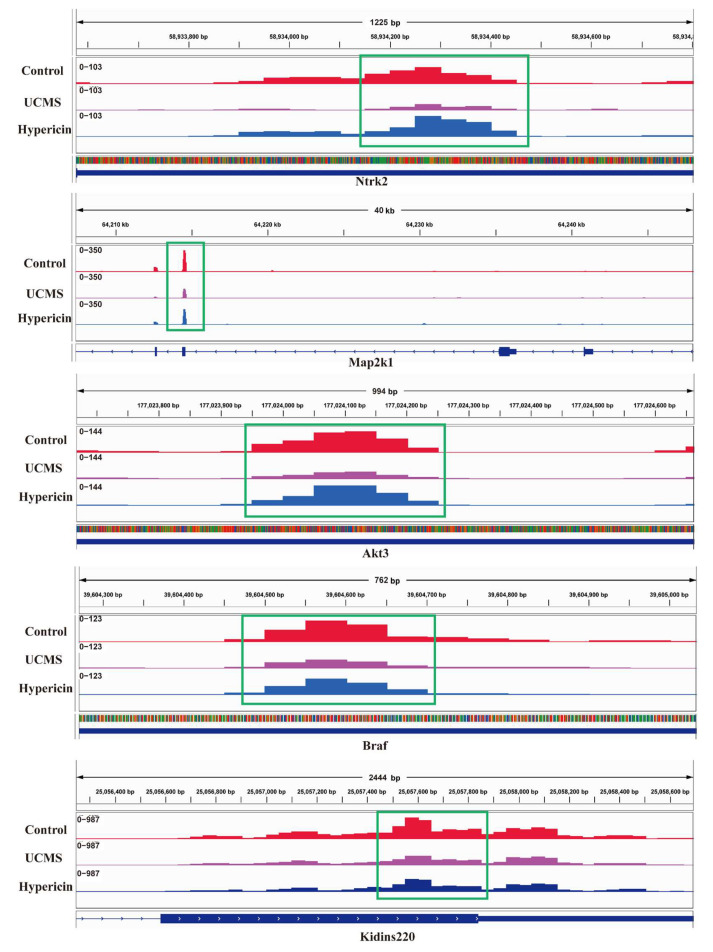
Abundant m6A in Akt3, Ntrk2, Braf, Map2k1, and Kidins220 mRNA transcripts in control, UCMS, and hypericin treatment.

**Table 1 molecules-28-03859-t001:** DMPs enriched in neurotrophin signaling pathway.

Genes	Peak Start	Peak End	UCMS vs. Control		UCMS vs. Hypericin	
			*p*-Value	log2FC	*p*-Value	log2FC
NTRK2	58,933,914	58,934,464	8.71 × 10^−3^	−0.868	5.00 × 10^−2^	−0.644
MAP2K1	64,214,536	64,240,811	1.32 × 10^−5^	−1.36	2.45 × 10^−3^	−0.806
AKT3	177,024,015	177,024,315	3.16 × 10^−2^	−1.12	4.68 × 10^−2^	−1.14
BRAF	39,604,527	39,604,777	1.15 × 10^−2^	−1.16	1.62 × 10^−4^	−1.72
KIDINS220	25,056,695	25,058,240	8.51 × 10^−4^	−0.914	8.91 × 10^−3^	−0.645

**Table 2 molecules-28-03859-t002:** Genes of RNA-seq enriched in neurotrophin signaling pathway.

Genes	Description	UCMS vs. Control		UCMS vs. Hypericin	
		*p*-Value	log2FC	*p*-Value	log2FC
Sort1	Sortilin 1	0.04426	0.14	0.04656	0.24
Camk4	Calcium/calmodulin dependent protein kinase IV	0.00635	−0.27	0.04506	−0.24
Arhgdig	Rho GDP dissociation inhibitor gamma	0.02523	−0.24	0.02066	−0.36

## Data Availability

Not applicable.

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
