# Peer review of "Hypericin Ameliorates Depression-like Behaviors via Neurotrophin Signaling Pathway Mediating m6A Epitranscriptome Modification"

_molecules, 2023, doi:10.3390/molecules28093859_

Round 1

Reviewer 1 Report

In this study by Lei et al., entitled “Hypericin ameliorates depression-like behaviors via Neurotrophin signaling pathway mediating m6A epitranscriptome modification”, the authors investigated the epitranscriptomic changes after hypericin treatment in a USCS mice model. First, the authors showed the antidepressant effect of hypericin using several behavioral tests. Next, the authors explore epitranscriptomic mechanisms of antidepressant effect by focusing on the gene and protein expression changes of m6A regulators. They further extended the study by conducting m6A sequencing and RNA sequencing. They found the enrichment of differentially methylated or differentially expressed genes in a gene ontology term related to the neurotrophin signaling pathway. While their studies add new support to the epitransciptomic mechanism of the antidepressant effect, detailed descriptions of data and analysis are much needed to support their conclusions.

1.       Adequate references need to be included in the Introduction. The authors should insert the corresponding references whenever stating evidence from the literature.

2.       In section 3.2, did you check gene expression changes of m6A writers, erasers, and readers in your RNA-seq data? Does the result look similar to the qPCR results?

3.       The authors should remake bar graphs by incorporating dot plots and showing the values of each subject.

4.       In section 3.2, did you check the gene expression of other m6A writers and readers? The authors should state why METTL3, METTL14, WTAP, FTO, and ALKBH5 were selected as targets.

5.       In the 2.5 MeRIP sequencing section, the authors should include a detailed procedure for m6A immunoprecipitation. For example, which antibody was used?

6.       In the 2.6 MeRIP-qPCR section, was the total RNA input and antibody ratio the same as the MeRIP-sequencing preparation?

7.       In Figure 3B, the p-value of GGAC enrichment is small compared to the literature. Was the GGAC motif the top motif?

8.       The authors showed enrichment of differentially methylated genes and differentially expressed genes in GO terms related to neurotrophic factors. However, the authors didn’t show an overlap between the two analyses.

9.       The authors provided primer designs used for MeRIP-qPCR. Do these locations correspond to the m6A peak found in differential peak analysis?

10.     In 307-309, the authors should rewrite a sentence. Engel et al., (2018) found gene expression changes of METTL3 in different brain regions, not in the hippocampus.

11.     In line 140, the authors mentioned, “Differential peak analysis was performed with MeTPeak [11]”. However, in line 212, the authors mentioned, “The differentially methylated peaks (DMPs) were identified by MeTdiff software ….”. The authors should use the same differential methylation analysis package in the whole analysis since two packages could give different results.

Author Response

  1. Adequate references need to be included in the Introduction. The authors should insert the corresponding references whenever stating evidence from the literature.

Response: Thanks for your suggestions. We inserted the corresponding references, which have been highlighted in the revised manuscript. The revision part will be listed as follows:

Epigenetic modifications of genes play an important role in the development of depression [7-9], and similar to DNA methylation modifications and histone modifications, changes in the homeostasis of m6A modifications may also contribute to depression [10,11]. m6A modifications are among the most common modifications on RNA [12], and this dynamic and reversible modification depends mainly on regulation by methyltransferases (METTL3, METTL14, WTAP) and demethylases (FTO, ALKBH5) [11,13]. In eukaryotic organisms, m6A modifications are involved in biological processes by affecting RNA exit, shearing, translation, and degradation. Stable m6A modifications contribute to brain development [12,14]. Conversely, disruption of m6A modification usually leads to the development of various diseases, including depression.

The occurrence of depression is closely related to the disruption of m6A modification of RNA caused by changes in the activity or expression of m6A-modifying enzymes [10,11,15,16]. Demethylase FTO expression has been found to be markedly downregulated in the blood of depressed patients and in the hippocampus of mice with depressive-like behavior [15].

Another methyltransferase, METTL3, has also been associated with depression. Deletion of METTL3 in the hippocampus of mice can lead to decreased neurogenesis and depressive-like behaviors [18]. In addition, the antidepressant efficacy of tricyclic antidepressants (TCAs) has been found to be associated with m6A modifications [19].

  1. In section 3.2, did you check gene expression changes of m6A writers, erasers, and readers in your RNA-seq data? Does the result look similar to the qPCR results?

Response: Thanks. Yes, we checked those genes expression changes of m6A writers, erasers, and readers mRNA in the RNA-seq data. The results have shown that the mRNA expression dysregulation including WTAP and ALKBH5, etc. in the RNA-seq data are consistent to the results of qPCR.

  1. The authors should remake bar graphs by incorporating dot plots and showing the values of each subject.

Response: Thanks. We have replaced all the bar graphs in the article with dot plots (including figure 1, figure 2, figure4 and figure 5).

  1. In section 3.2, did you check the gene expression of other m6A writers and readers? The authors should state why METTL3, METTL14, WTAP, FTO, and ALKBH5 were selected as targets.

Response: Thanks. Yes. As the reviewer suggested, we checked the gene expression of other m6A writers and readers against the results of RNA-seq. The results of other m6A writers and readers expression have shown that the expression of YTHDF3, YTHDC1, and KIAA1429 was upregulated in mouse hippocampal tissue after UCMS stimulation but reversed by hypericin administration while the expression of YTHDF1 as well as YTHDC1 was decreased in the UCMS group but increased after hypericin administration. The genes of METTL3, METTL14, WTAP, FTO, and ALKBH5 were selected because those five genes are core components of the methyltransferase complex (METTL3, METTL14, WTAP) and major demethylases (FTO and ALKBH5). In addition, previous studies have reported that these genes involved into the pathogenesis of depression. Thus, we have made additional explanations in the article on line 115-118, which have been listed as follows,

Considering that METTL3, METTL14, and WTAP are core components of the methyltransferase complex, FTO and ALKBH5 as major demethylases, and also referred as the previously published studies on the depression pathogenesis, those five genes were selected as potential targets.  

  1. In the 2.5 MeRIP sequencing section, the authors should include a detailed procedure for m6A immunoprecipitation. For example, which antibody was used?

Response: Thanks. We added the experimental detailed procedure of m6A immunoprecipitation on line 390-399. In addition, we have expanded more detailed experimental steps in Supplementary materials. The revised part is listed as follows:

The m6A RNA Enrichment Kit (Epigentek, USA) was used to perform MeRIP assays according to the manufacturer's protocols. We incubated 10 ug of total mRNA with immunocapture buffer containing Immuno Capture Buffer, 2 ug of m6A antibody (Synaptic Systems), and affinity protein G beads for 90 minutes at room temperature under rotation so that the RNA with m6A modification binds to the affinity protein G beads and forms immunoprecipitation complexes, followed by the cleaved step with Nuclear Digestion Enhancer and Cleavage Enzyme Mix. The immunoprecipitation complex was then washed three times with wash buffer and then digested with proteinase K for 15 minutes at 55°C to release the enriched RNA from the magnetic beads. To purify and recover the enriched RNA, RNA binding particle analysis was performed.

  1. In the 2.6 MeRIP-qPCR section, was the total RNA input and antibody ratio the same as the MeRIP-sequencing preparation?

Response: Thanks. Yes, the total RNA and m6A antibody in the MeRIP-qPCR are consistent with the immunoprecipitation in MeRIP-seq (10 ug total RNA and 2 ug m6A antibody). We added in the line 427-429. Additional content is as follows:

The 10ug total RNA were input and 2 ug m6A antibody were immunoprecipitated in the MeRIP-qPCR, which are consistent with the MeRIP-seq.

  1. In Figure 3B, the p-value of GGAC enrichment is small compared to the literature. Was the GGAC motif the top motif?

Response: Thanks. Yes, we examined that the GGAC motif was one of the top significant motifs.

  1. The authors showed enrichment of differentially methylated genes and differentially expressed genes in GO terms related to neurotrophic factors. However, the authors didn’t show an overlap between the two analyses.

Response: Thanks. We supplemented the overlap of biological function enrichment analysis between the differentially methylated genes and differentially expressed genes on line 204-208. Additional content is as follows:

We further examined the overlapping biological terms in the gene enrichment analysis of differentially methylated genes in the MeRIP-seq and differentially expressed genes in the mRNA-seq, the four important pathways including apelin signaling pathway, aldosterone synthesis and secretion, RNA degradation and neurotrophin signaling pathway were found to be shared with these two enrichment analysis.

  1. The authors provided primer designs used for MeRIP-qPCR. Do these locations correspond to the m6A peak found in differential peak analysis?

Response: Thanks. Yes, the primers of MeRIP-qPCR were designed according to the identified m6A peak in differential peak analysis. We have added a note in the article on line 431-432. Additional content is as follows:

The primers of MeRIP-qPCR were designed according to the identified m6A peaks in differential peak analysis.

  1. In 307-309, the authors should rewrite a sentence. Engel et al., (2018) found gene expression changes of METTL3 in different brain regions, not in the hippocampus.

Response: Thanks. We rewritten the sentence on line 264-266. The modifications are as follows:

Mareen Engel et al. (2018) found the gene expression of METTL3 was downregulated in the medial prefrontal cortex and the central amygdala of the stressed mice [31].

  1. In line 140, the authors mentioned, “Differential peak analysis was performed with MeTPeak [11]”. However, in line 212, the authors mentioned, “The differentially methylated peaks (DMPs) were identified by MeTdiff software ….”. The authors should use the same differential methylation analysis package in the whole analysis since two packages could give different results.

Response: Thanks. In the study, MeTPeak software not MeTdiff were used to analysis and we have corrected it.

Reviewer 2 Report

Reviewer does not feel qualified to evaluate the methods used for correctness of application and implementation. Also, this reviewer cannot properly determine some of the results. This is especially true of the results from sections 3.3 through 3.5. The original blots images from the supplementary file were also not assessable by me. Reviewer therefore wishes to limit himself here to some general comments. The English language used by the authors in the manuscript needs improvement. The abbreviation i.g. cannot be brought home. What is shown in Figures 3, 4 and supplementary figures is not sufficiently clear to referent. The supplementary tables are useful, however.

In general, the manuscript cannot be read by someone not well versed in the molecular biological techniques used. The introduction and discussion offer little support in this regard because virtually nothing is explained in them: neither about the methods used nor the interpretation of the findings. The abstract is also unclear. It mentions "MeRIP-seq and mRNA-seq experiments" without allowing the unprepared reader to have any idea what most of the article is about. The conclusions are largely out of whack in the abstract and discussion. The text therefore fits better with a report summary of research findings than an inspirational article. Referee is aware that this journal also calls for mentioning all the details, but it would have been better to describe them in the supplementary material and to explain better in the article why one investigates all this and what conclusions may be drawn from the various steps.

Reference also has some general comments. (1) The FTO gene has been studied primarily in connection with the development of obesity. This effect may also come about by other means than through bringing about m6A methylation modifications (doi: 10.2147/PGPM.S327353). 2. Hypericin affects cerebral CYP enzymes and serotonin homeostasis. Serotonergic innervation of the hippocampus plays an important role in the development of depressive responses. 3. The authors examine changes in the hippocampus of mice, why was this tissue chosen and not the habenula, for example? The hippocampus probably plays a major role in remembering the context of stress-inducing conditions and is connected to the habenula via the hypothalamus and septal nuclei. 4. The sucrose preference test, while widely used for to model anhedonia, probably also says something (perhaps even much more) about satiety mechanisms in the hypothalamus. This does have relevance in connection with the link of the experiments with the FTO gene. 5. The pharmacological profile of duloxetine is not compared with that of hypericin, nor are the results interpreted in that context. 6. Why was no active control (duloxetine) included in a significant portion of the molecular pharmacological experiments and the results are not commented on from that perspective?

Author Response

  1. The English language used by the authors in the manuscript needs improvement.

Response: Thanks for your suggestions. We improved English language in the manuscript.

  1. The abbreviation i.g. cannot be brought home.

Response: Thanks. We used oral administration instead of abbreviation i.g. on line 348 and 349.

  1. What is shown in Figures 3, 4 and supplementary figures is not sufficiently clear to referent. The supplementary tables are useful, however.

Response: Thanks for your suggestions. We have adjusted the resolution of Figures 3, 4 and supplementary figures to make the images more clearly displayed.

  1. In general, the manuscript cannot be read by someone not well versed in the molecular biological techniques used. The introduction and discussion offer little support in this regard because virtually nothing is explained in them: neither about the methods used nor the interpretation of the findings.

Response: We highly appreciated this comment. We modified the introduction by adding in the biological background about m6A modification, and supplemented the description of m6A modification in the pathology of depression and antidepressants. Meanwhile, we described the detection methods of m6A modification and its advantages for the MeRIP-seq. For the discussion parts, we have made major revision according to the reviewer suggestions, we discussed the results and findings in the different experiments. The revised parts are listed as follows:

For introduction, on line 38-90. The modified sentences have been shown with underlined text.

Major depressive disorder (MDD) is one of the most severe mental illnesses with prevalence rates ranging from 2.2% to 26.8% of the world [1,2]. According to the World Health Organization (WHO), more than 800,000 people die by suicide each year due to depression [1,3]. But, the origin of depression is not clearly understood. In recent decades, the serotonin imbalance theory has been widely accepted, and SSRIs have been developed as antidepressants [4,5]. However, recent studies have challenged the serotonin imbalance theory and indicated that not only is there no direct relationship between serotonin activity and depression, but depressed patients may develop a life-long dependence on the drug based on the serotonin imbalance theory [6]. Therefore, it is necessary to explore new molecular mechanisms in the pathogenesis of depression and to develop new antidepressants. Epigenetic modifications of genes play an important role in the development of depression [7-9], and similar to DNA methylation modifications and histone modifications, changes in the homeostasis of m6A modifications may also contribute to depression [10,11]. m6A modifications are among the most common modifications on RNA [12], and this dynamic and reversible modification depends mainly on regulation by methyltransferases (METTL3, METTL14, WTAP) and demethylases (FTO, ALKBH5) [11,13]. In eukaryotic organisms, m6A modifications are involved in biological processes by affecting RNA exit, shearing, translation, and degradation. Stable m6A modifications contribute to brain development [12,14]. Conversely, disruption of m6A modification usually leads to the development of various diseases, including depression.

The occurrence of depression is closely related to the disruption of m6A modification of RNA caused by changes in the activity or expression of m6A-modifying enzymes [10,11,15,16]. Demethylase FTO expression has been found to be markedly downregulated in the blood of depressed patients and in the hippocampus of mice with depressive-like behavior [15]. Also, targeted suppression of FTO expression in the hippocampus of mice showed depressive-like behaviors in behavioral tests, whereas over-expression of FTO mRNA showed antidepressant effects. Reduction of FTO activity directly increased m6A methylation modifications at Adrb2 mRNA, decreased Adrb2 mRNA stability, and altered the expression of c-MYC and Sirt1 in downstream signaling pathways [15]. Chronic stress was reported to impair hippocampal synaptic plasticity in mice while decreasing protein levels of FTO [17]. Another methyltransferase, METTL3, has also been associated with depression. Deletion of METTL3 in the hippocampus of mice can lead to decreased neurogenesis and depressive-like behaviors [18]. In addition, the antidepressant efficacy of tricyclic antidepressants (TCAs) has been found to be associated with m6A modifications [19]. Overall, these studies have provided some impressive evidence for a link between epitranscriptomic modifications, depressive-like behaviors, and antidepressant effects.

As an important natural antidepressant, St. John's wort is widely used in the treatment of patients with mild to moderate depression because this antidepressant has good clinical efficacy and fewer side effects than SSRIs [20,21]. However, the mechanism of antidepressant action remains unclear. In addition, a recent study has shown that hyperforin, the active ingredient in St. John's wort extract, inhibits the endothelial-to-mesenchymal transition by mediating the methyltransferase METTL3 [22]. We speculate that the antidepressant effect of hypericin may be related to m6A modification of epitranscriptomic regulation; however, no such studies have been published. In the present study, we established an unpredictable chronic mild stress and orphan mouse model to investigate whether the antidepressant efficacy of hypericin. The expression of five classical methylation-modifying enzymes (METTL3, METTL14, WTAP, FTO, ALKBH5) was examined using molecular pharmacology assays including MeRIP-seq which can detect genome-wide m6A modification changes in the hippocampus by capturing RNA fragments with m6A antibodies and combining second-generation sequencing, and followed by bioinformatic analysis to investigate the antidepressant mechanisms of hypericin in combination with the transcriptome data.

Discussion, on line 233-319. The modified sentences have been shown with underlined text.

 In our study, we investigated the antidepressant efficacy of hypericin by establishing a UCMS mouse model in combination with behavioral tests, using duloxetine as a positive control for the antidepressant evaluation of hypericin and focusing on the antidepressant mechanism of hypericin. The results of TST and FST showed that treatment with hypericin and duloxetine significantly reduced the time to giving up struggle in a despairing environment in UCMS mice, and the results of SPT demonstrated that treatment with hypericin and duloxetine restored the pleasure deficit in UCMS mice. In addition, SPT also reflected hypothalamic satiety mechanisms [23], suggesting that the occurrence of depression and the efficacy of antidepressants may be related to obesity. The studies based on clinical data have found a negative correlation between depressive symptoms and body weight in humans [24], and SSRIs treatment has also been shown to be associated with weight gain in depressed patients [25,26]. To be interesting, obesity was also found to be regulated by the FTO gene. Recent genetic studies found that four polymorphic variants in the FTO gene were associated with body mass index in schizophrenia patients [27]. Meanwhile, FTO, as one important m6A demethylase, was found that knock out FTO mice have shown depressive like behavior and that FTO expression may correlate with the antidepressant efficacy of fluoxetine [15]. These studies suggested the therapeutic effect of hypericin may be associated with m6A methylation modifications and obesity. In present study, OFT test was performed as a complement behavior test to measure depressive behavior of UCMS model, despite that the OFT was commonly used to detect anxiety-like behavior [28]. From the results from OFT, the spontaneous activity exploration behavior of UCMS mice increased significantly after treatment with hypericin and duloxetine. In conclusion, hypericin has comparable antidepressant efficacy to duloxetine.

The expression of multiple m6A enzymes were measured in the hippocampus of mice. Hippocampal tissue, as one of the central brain regions, has connections to the habenula via the hypothalamus and septal nuclei and involve into the stress regulation [29,30]. We found that mRNA and protein levels of the methyltransferase METTL3, WTAP, decreased in the hippocampus after UCMS stimulation in mice, and hypericin administration significantly increased the expression of both. METTL3, an important component of methyltransferases, is involved in several biological processes in the brain, including memory formation and consolidation, synaptic plasticity, and neural stem cell differentiation [31,32]. Recently, Mareen Engel et al. (2018) found the gene expression of METTL3 was downregulated in the medial prefrontal cortex and the central amygdala of the stressed mice [31]. METTL3 deficient mice were exhibited reduced hippocampal neurogenesis, which contributes to depressive behavior [18]. These results suggested that the antidepressant effect of hypericin may be related to altered expression of METTL3 and WTAP.

After finding that hypericin increased the expression of methyltransferase METTL3 and WTAP in the hippocampus of UCMS mice, we examined the changes in m6A modification and mRNA expression in mouse hippocampal tissue on a genome-wide scale. The results of MeRIP-seq and RNA-seq exhibited that after UCMS stimulation, the homeostasis of m6A modification in mouse hippocampus was disturbed and the expression levels of some mRNAs changed. After hypericin treatment, the homeostasis of m6A modification was regulated and mRNA expression levels were modulated. Subsequent biological functional enrichment analysis showed that the various mRNAs and the DEGs were enriched in the apelin signaling pathway, aldosterone synthesis and secretion, RNA degradation, and neurotrophin signaling pathway. The neurotrophin signaling pathway included four types of neurotrophins, including nerve growth factor (NGF), brain-derived neurotrophic factor (BDNF), neurotrophin-3 (NT-3), and neurotrophin-4 (NT-4) [33,34], its biological function was related to regulate neuronal development including synapse formation and synaptic plasticity [35,36]. More and more evidence has suggested that it mediated into the onset of depression and many antidepressants act [37-39].

DEGs (Akt3, Ntrk2, Braf, Map2k1, and Kidins220) and differential genes (Sort1, Camk4, and Arhgdig) were extracted from the neurotrophin signaling pathway. In conjunction with subsequent validation, we found that in mice stimulated with UCMS, m6A modification of Akt3, Ntrk2, Braf, and Kidins220 mRNAs decreased after UCMS stimulation and increased after hypericin treatment. With respect to Ntrk2, we confirmed that m6A methylation and mRNA expression of Ntrk2 were reduced in the hippocampus of UCMS mice and that expression was reversed after hypericin treatment. As a receptor for BDNF, Ntrk2 is key to the efficacy of many antidepressants [40,41]. It has been suggested that combined treatment with fluoxetine and melatonin may have a synergistic antidepressant effect by restoring BDNF-Ntrk2 signaling in the hippocampus [42]. In addition, the antidepressant efficacy of (R)-ketamine was also found to be strongly associated with Ntrk2 [43]. In conclusion, the dysregulated expression of Ntrk2 played the contributory role on the depression and the target of novel antidepressants.  

In addition to genes enriched in the neurotrophic factor signaling pathway, we also focused on a number of other genes with large differential variation. We found that dysregulated expression of the mRNAs of Shank1, Syt7, and Igfbp5 could be associated with depressive behavioral phenotypes. Interestingly, the methylation modification of Shank1 was also significantly altered in the UCMS mice. It has been reported that Shank1, as a gene associated with neuroplasticity, was decreased in the medial prefrontal cortex and hippocampus of mice with depressive behavior [44,45]. Although these changes in gene expression could not be reversed by hypericin, they are closely related to the onset of depression and could be potential targets for antidepressants.

Apart from the m6A modification mechanisms related into hypericin antidepressant effect, it has also found that hypericin can potently inhibit the activities of P450 enzymes (CYP) in vitro experiment [46] and modify the levels of dopamine and serotonin in the synaptic clefts [47,48]. Recent studies have also shown that hypericin can exert antidepressant effects via inhibiting neuroinflammation. Zhai et al (2022) found that hypericin can effectively alleviate the symptoms of postpartum depression in rats by inhibiting NLRP3 inflammasome activation and regulating glucocorticoid metabolism [49]. Another study also reported that chlorogenic acid and hypericin can exert antidepressant effects via the gut microbiota-neuroinflammatory axis [50]. These above studies suggested that hypericin may produce antidepressant effects through multiple potential mechanisms. However, it is unclear whether the expression of key genes and molecular targets in those hypericin related mechanisms are regulated by m6A modifications, which warrants further investigation.

  1. The abstract is also unclear. It mentions "MeRIP-seq and mRNA-seq experiments" without allowing the unprepared reader to have any idea what most of the article is about. The conclusions are largely out of whack in the abstract and discussion. The text therefore fits better with a report summary of research findings than an inspirational article.

Response: Thanks for this suggestion. We largely revised the abstract and clarified the related experimental procedures, and supplemented the results of each experiment to make it the consistent with discussion. Meanwhile, we described the meanings about the novel mechanisms on antidepressant effect of hypericin on line 11-33. The revised parts are listed as follows:

Hypericin, one of the major antidepressant constituents of St. John's wort, has been shown to exert antidepressant effects by affecting cerebral CYP enzymes, serotonin homeostasis, and neuroinflammatory signaling pathways, but the exact mechanism is unknown. Previous clinical studies reported that the mRNA modification N6-methyladenosine (m6A) interferes with the neurobiological mechanism in depressed patients, and it was also found that the antidepressant efficacy of tricyclic antidepressants (TCAs) is related to m6A modifications. Therefore, we hypothesize that the antidepressant effect of hypericin may be related to m6A modification of epitranscriptomic regulation. We constructed a UCMS mouse depression model and found that hypericin ameliorated depressive-like behavior in UCMS mice. Molecular pharmacology experiments showed that hypericin treatment upregulated the expression of m6A-modifying enzymes METTL3 and WTAP in the hippocampus of UCMS mice. Next, we performed MeRIP-seq and RNA-seq to study m6A modifications and changes in mRNA expression on a genome-wide scale separately. The genome-wide m6A assay and MeRIP-qPCR results showed that the m6A modifications of Akt3, Ntrk2, Braf, and Kidins220 mRNA were significantly altered in the hippocampus of UCMS mice after stress stimulation and were reversed by hypericin treatment. Transcriptome assays and qPCR results showed that the Camk4 and Arhgdig genes might be related to the antidepressant efficacy of hypericin. Further gene enrichment results showed that the differential genes were mainly involved in neurotrophic factor signaling pathways. In conclusion, our results show that hypericin upregulates m6A methyltransferase METTL3 and WTAP in the hippocampus of UCMS mice and stabilizes m6A modifications to exert antidepressant effects via the neurotrophin signaling pathway. This suggests that METTL3 and WTAP-mediated changes in m6A modifications may be a potential mechanism for the pathogenesis of depression and the efficacy of antidepressants, and that neurotrophin signaling pathway plays a key role in this process.

  1. Referee is aware that this journal also calls for mentioning all the details, but it would have been better to describe them in the supplementary material and to explain better in the article why one investigates all this and what conclusions may be drawn from the various steps.

Response: Thanks for your suggestions. We added all the experimental details in the supplementary material. And we elaborated the purpose of each experiment and the conclusions drawn in the results part of article in the results parts, which have highlighted in the revised parts.

  1. The FTO gene has been studied primarily in connection with the development of obesity. This effect may also come about by other means than through bringing about m6A methylation modifications (doi: 10.2147/PGPM.S327353).

Response: Thanks. As suggested by reviewer, we added the description and literature on the FTO gene connection with obesity (line 239-250). The revised parts are listed as follows:

In addition, SPT also reflected hypothalamic satiety mechanisms [23], suggesting that the occurrence of depression and the efficacy of antidepressants may be related to obesity. The studies based on clinical data have found a negative correlation between depressive symptoms and body weight in humans [24], and SSRIs treatment has also been shown to be associated with weight gain in depressed patients [25,26]. To be interesting, obesity was also found to be regulated by the FTO gene. Recent genetic studies found that four polymorphic variants in the FTO gene were associated with body mass index in schizophrenia patients [27]. Meanwhile, FTO, as one important m6A demethylase, was found that knock out FTO mice have shown depressive like behavior and that FTO expression may correlate with the antidepressant efficacy of fluoxetine [15]. These studies suggested the therapeutic effect of hypericin may be associated with m6A methylation modifications and obesity.

  1. Hypericin affects cerebral CYP enzymes and serotonin homeostasis. Serotonergic innervation of the hippocampus plays an important role in the development of depressive responses.

Response: Thanks for your suggestions. As reviewer mentioned, previous studies have shown that the antidepressant efficacy of hypericin is related into the activities cerebral CYP enzymes and serotonin homeostasis. We have added a discussion of the relevant literature in the discussion section on line 307-310. The revised parts are listed as follows:

 Apart from the m6A modification mechanisms related into hypericin antidepressant effect, it has also found that hypericin can potently inhibit the activities of P450 enzymes (CYP) in vitro experiment [46] and modify the levels of dopamine and serotonin in the synaptic clefts [47,48].

  1. The authors examine changes in the hippocampus of mice, why was this tissue chosen and not the habenula, for example? The hippocampus probably plays a major role in remembering the context of stress-inducing conditions and is connected to the habenula via the hypothalamus and septal nuclei.

Response: Thanks. As reviewer have mentioned, the reason for selection hippocampus was considered that hippocampal tissue, as one of the central brain regions, has connections to the habenula via the hypothalamus and septal nuclei and involve into the stress regulation. We have also made additional explanations in the article on line 256-259. The revised parts are listed as follows:

The expression of multiple m6A enzymes were measured in the hippocampus of mice. Hippocampal tissue, as one of the central brain regions, has connections to the habenula via the hypothalamus and septal nuclei and involve into the stress regulation [29,30].

  1. The sucrose preference test, while widely used for to model anhedonia, probably also says something (perhaps even much more) about satiety mechanisms in the hypothalamus. This does have relevance in connection with the link of the experiments with the FTO gene.

Response: Thanks for your suggestions. Yes, the sucrose preference test can be used to detect not only anhedonia but also the influence of other regulatory mechanisms (satiety mechanisms in the hypothalamus). We added and discussed relevant literature to the article on line 239-250. The revised parts are listed as follows:

In addition, SPT also reflected hypothalamic satiety mechanisms [23], suggesting that the occurrence of depression and the efficacy of antidepressants may be related to obesity. The studies based on clinical data have found a negative correlation between depressive symptoms and body weight in humans [24], and SSRIs treatment has also been shown to be associated with weight gain in depressed patients [25,26]. To be interesting, obesity was also found to be regulated by the FTO gene. Recent genetic studies found that four polymorphic variants in the FTO gene were associated with body mass index in schizophrenia patients [27]. Meanwhile, FTO, as one important m6A demethylase, was found that knock out FTO mice have shown depressive like behavior and that FTO expression may correlate with the antidepressant efficacy of fluoxetine [15]. These studies suggested the therapeutic effect of hypericin may be associated with m6A methylation modifications and obesity.

  1. The pharmacological profile of duloxetine is not compared with that of hypericin, nor are the results interpreted in that context. Why was no active control (duloxetine) included in a significant portion of the molecular pharmacological experiments and the results are not commented on from that perspective?

Response: Thanks. In present study, we focused on the antidepressant mechanism of hypericin, and duloxetine was included as a positive control for drug effect evaluation. Therefore, we did not explore the antidepressant mechanism of duloxetine with molecular pharmacological experiments.  We have made additional explanations in the article on line 233-236. The revised parts are listed as follows:

In our study, we investigated the antidepressant efficacy of hypericin by establishing a UCMS mouse model in combination with behavioral tests, using duloxetine as a positive control for the antidepressant evaluation of hypericin and focusing on the antidepressant mechanism of hypericin.

Reviewer 3 Report

This paper presents the results of a study examining the effects of hypericin - the main active ingredient of St John's wort (SJW), which is already in use as an antidepressant - on a specific molecular pathway considered to be of importance in depression (m6A modification), using a chronic unpredictable stress model of depression in male mice.

While hypericin (or rather SJW) is already widely used in several countries for the treatment of depression, it is true that its exact mechanism of action remains unclear. Early research examined the action of this drug on monoaminergic transmission (particularly serotonin). However, there have been subsequent investigations of other molecular pathways relevant to the antidepressant actions of SJW, such as immune-inflammatory processes. The current study has merit in that it attempts to examine a novel mechanism that could both underlie the action of hypericin and lead to the development of new antidepressants that target this mechanism.

The paper could be improved by attention to the following details:

1. The introductory section on the epidemiology of depression is unsatisfactory. 3.8% is a low estimate for the global prevalence of depression (unless the authors are quoting an unadjusted point prevalence estimate). It would be more useful to provide a range (based on large-scale epidemiological surveys, such as the World Mental Health Survey) instead of a single figure.

2. The statement that 800,000 people "die by MDD" every year is inaccurate. The authors should specify whether they are referring to suicides related to MDD (where there is a direct causal link) or whether they are referring to deaths caused by general medical conditions (cardiac, immune) where depression is thought to play a role. MDD per se does not lead to death, though it can be a risk factor for death by suicide, accident or comorbid medical illness.

3. The transition between the epidemiological and basic science sections of the introduction ("More and more evidences..." [sic]) is abrupt. The authors could introduce the key areas of focus of their paper more organically by mentioning the limitations of existing models of depression (for example, a recent paper by Moncrieff et al. 2022 critiques the popular "serotonin hypothesis") and then highlight the need to identify novel molecular mechanisms related to depression. This could be followed by the discussion of in vitro, animal and human research linking m6A modification to depression.

4. The statement that SJW is a "first-line antidepressant" is somewhat questionable. While it is widely used, it is often used as a second-line option or an alternative treatment in patients who do not tolerate conventional antidepressants. It would be better to use a more neutral term, such as "widely used" or "of proven efficacy".

5. In the "Materials and Methods" section, please provide accurate details of how many mice were used for the study, and how many were in the "UCMS" and "control" groups. The spelling of "Duloxetine" should also be corrected in this paragraph.

6. The authors have mentioned using the Open Field Test in their experimental animals. As this is usually used to measure anxiety rather than depression, the authors should provide some justification for its use in their protocol (e.g., are they trying to demonstrate both anti-depressant and anti-anxiety effects related to m6A modification?)

7. The section on statistical analysis (lines 168-171) is too brief for a study of this sort. Are t-tests / analyses of variance sufficient to identify meaningful associations, given the wealth of data generated? What corrections / precautions did the researchers adopt when analyzing the data to minimize the risk of false-positive associations?

8. What was the rationale for the inclusion of a duloxetine control group? Duloxetine is known to work as an inhibitor of both serotonin and noradrenaline uptake. Are the authors also examining its effects on m6A modification? If not, is it necessary to introduce an added layer of complexity to the analysis and interpretation of the results by including this drug?

9. Did the authors examine the effects of duloxetine on gene expression / methylation? If not, it is not possible to conclude that the effects observed are specific to hypericin alone - they could be a general effect observed with several antidepressants. No results related to duloxetine (except on behavioural measures) have been presented in the paper.

10. The discussion focuses exclusively on the study findings and on the m6A modification pathway. While I agree that this is the focus of the paper, the authors should also discuss how their findings fit into the existing research in this field, particularly pertaining to hypericin. (For example, Zhai et al. found an effect of hypericin on the NLRP3 inflammasome, which is also considered to be of importance in depression.) How can the current study's results be integrated into the "broader picture" of what we understand about the molecular mechanisms underlying depression?

11. The Conclusion section should not just repeat the study results; it should provide a summary of key study findings (positive and negative) and discuss their implications for research and practice.

12. A substantial degree of language editing is needed to address errors in spelling ("duloxetine" for "doluxetine", "relieve" for "relive") and grammar.

Author Response

  1. The introductory section on the epidemiology of depression is unsatisfactory. 3.8% is a low estimate for the global prevalence of depression (unless the authors are quoting an unadjusted point prevalence estimate). It would be more useful to provide a range (based on large-scale epidemiological surveys, such as the World Mental Health Survey) instead of a single figure.

Response: Thanks for your suggestions. We revised the epidemiology of depression on line 38-39. The revised parts are listed as follows:

Major depressive disorder (MDD) is one of the most severe mental illnesses with prevalence rates ranging from 2.2% to 26.8% of the world [1,2].

  1. The statement that 800,000 people "die by MDD" every year is inaccurate. The authors should specify whether they are referring to suicides related to MDD (where there is a direct causal link) or whether they are referring to deaths caused by general medical conditions (cardiac, immune) where depression is thought to play a role. MDD per se does not lead to death, though it can be a risk factor for death by suicide, accident or comorbid medical illness.

Response: Thanks. We have corrected our statement to “more than 800,000 people who died by suicide each year because of depression” on line 39-41. The revised parts are listed as follows:

According to the World Health Organization (WHO), more than 800,000 people die by suicide each year due to depression [1,3].

  1. The transition between the epidemiological and basic science sections of the introduction ("More and more evidences..." [sic]) is abrupt. The authors could introduce the key areas of focus of their paper more organically by mentioning the limitations of existing models of depression (for example, a recent paper by Moncrieff et al. 2022 critiques the popular "serotonin hypothesis") and then highlight the need to identify novel molecular mechanisms related to depression. This could be followed by the discussion of in vitro, animal and human research linking m6A modification to depression.

Response: We greatly appreciate your comment. As suggested by the reviewer, we have changed the transition between the section on the epidemiological and basic science in the introduction on line 41-46. The revised parts are listed as follows:

Major depressive disorder (MDD) is one of the most severe mental illnesses with prevalence rates ranging from 2.2% to 26.8% of the world [1,2]. According to the World Health Organization (WHO), more than 800,000 people die by suicide each year due to depression [1,3]. But, the pathogenesis of depression is still unclear. In recent decades, the serotonin imbalance theory has been widely accepted, and SSRIs have been developed as antidepressants [4,5]. However, recent studies have challenged the serotonin imbalance theory and indicated that not only is there no direct relationship between serotonin activity and depression, but depressed patients may develop a life-long dependence on the drug based on the serotonin imbalance theory [6]. Therefore, it is necessary to explore new molecular mechanisms in the pathogenesis of depression and to develop new antidepressants.

  1. The statement that SJW is a "first-line antidepressant" is somewhat questionable. While it is widely used, it is often used as a second-line option or an alternative treatment in patients who do not tolerate conventional antidepressants. It would be better to use a more neutral term, such as "widely used" or "of proven efficacy".

Response: Thanks for your suggestions. Based on the reviewer suggestions, we have corrected the questionable statement on SJW as first-line antidepressant on line 75-77. The revised parts are listed as follows:

As an important natural antidepressant, St. John's wort is widely used in the treatment of patients with mild to moderate depression because this antidepressant has good clinical efficacy and fewer side effects than SSRIs [20,21].

  1. In the "Materials and Methods" section, please provide accurate details of how many mice were used for the study, and how many were in the "UCMS" and "control" groups. The spelling of "Duloxetine" should also be corrected in this paragraph.

Response: Thanks. We added the accurate details on line 346-350 (n = 8 in each group). Meanwhile, we corrected the spelling of “Duloxetine”. The revised parts are listed as follows:

After the three-week treatment, the mice in the UCMS group were randomly divided into three groups (n = 8 in each group): UCMS group, UCMS +duloxetine (duloxetine, 30 mg/kg, oral administration) group, and UCMS +hypericin (hypericin, 2.4 mg/kg, oral administration) group.

  1. The authors have mentioned using the Open Field Test in their experimental animals. As this is usually used to measure anxiety rather than depression, the authors should provide some justification for its use in their protocol (e.g., are they trying to demonstrate both anti-depressant and anti-anxiety effects related to m6A modification?)

Response: Thanks. As the reviewer mentioned, the Open Field Test was mainly used to detect anxiety-like behavior. In the Open Field Test, the total distance of mice movement also reflects their ability to move autonomously which have used to measure the move ability of depression-like behavior mice. Thus, OFT as a complement behavior test was used to measure depressive behavior of UCMS model. We also added a note in the article on line 250-255. The revised parts are listed as follows:

In present study, OFT test was performed as a complement behavior test to measure depressive behavior of UCMS model, despite that the OFT was commonly used to detect anxiety-like behavior [28]. From the results from OFT, the spontaneous activity exploration behavior of UCMS mice increased significantly after treatment with hypericin and duloxetine.

  1. The section on statistical analysis (lines 168-171) is too brief for a study of this sort. Are t-tests / analyses of variance sufficient to identify meaningful associations, given the wealth of data generated? What corrections / precautions did the researchers adopt when analyzing the data to minimize the risk of false-positive associations?

Response: Thanks. As the reviewer suggested, we added statistical analysis methods, software, and correction methods in the article on line 448-461. The revised parts are listed as follows:

In this study, the number of animals for behavioral testing was determined based on our previous experimental experience (eight animals per group), and qPCR, MeRIP-qPCR and western blot were repeated at least three times independently. All data are calculated with the mean value and standard deviation. First, we used the Shapirp-Wilk normality test to check whether the individual data were normally distributed. Then, the Bartletts test was performed to ensure that the individual data had the same SDs. If the individual data followed a normal distribution and the group variances were homogeneous, we used the one-way ANOVA with Dunnett's post hoc test. When group variances were not homogeneous, we used the Brown-Forsythe and Welch ANOVA test with the Dunnett's post hoc test. All statistical analyses were performed using GraphPad Prism 9 software (GraphPad, USA). For histological data analysis, we utilized Fastqc, Hisat2, DESeq2, STAR aligner, ChIPseeker, Exomepeak2 and MeTPeak for data cleaning, quality control, genomic matching and differential analysis. The threshold of p < 0.05 was considered statistical significance.

  1. What was the rationale for the inclusion of a duloxetine control group? Duloxetine is known to work as an inhibitor of both serotonin and noradrenaline uptake. Are the authors also examining its effects on m6A modification? If not, is it necessary to introduce an added layer of complexity to the analysis and interpretation of the results by including this drug?

Response: Thanks. In present study, we focused on the antidepressant mechanism of hypericin, and duloxetine was included as a positive control for drug effect evaluation. Therefore, we did not attempt to investigate the relationship between the antidepressant efficacy of duloxetine and m6A modifications. We also added a note in the article on line 233-236. The revised parts are listed as follows:

In our study, we investigated the antidepressant efficacy of hypericin by establishing a UCMS mouse model in combination with behavioral tests, using duloxetine as a positive control for the antidepressant evaluation of hypericin and focusing on the antidepressant mechanism of hypericin.

  1. Did the authors examine the effects of duloxetine on gene expression / methylation? If not, it is not possible to conclude that the effects observed are specific to hypericin alone - they could be a general effect observed with several antidepressants. No results related to duloxetine (except on behavioural measures) have been presented in the paper.

Response: Thanks. In the present study, we did not examine the effects of duloxetine on gene expression / methylation. so, it is not possible to state that the observed results are specific to hypericin.

  1. The discussion focuses exclusively on the study findings and on the m6A modification pathway. While I agree that this is the focus of the paper, the authors should also discuss how their findings fit into the existing research in this field, particularly pertaining to hypericin. (For example, Zhai et al. found an effect of hypericin on the NLRP3 inflammasome, which is also considered to be of importance in depression.) How can the current study's results be integrated into the "broader picture" of what we understand about the molecular mechanisms underlying depression?

Response: Thanks. We added a discussion of relevant studies on the antidepressant effects of hypericin and explored the link between these results and our results on line 310-319. The revised parts are listed as follows:

Recent studies have also shown that hypericin can exert antidepressant effects via inhibiting neuroinflammation. Zhai et al (2022) found that hypericin can effectively alleviate the symptoms of postpartum depression in rats by inhibiting NLRP3 inflammasome activation and regulating glucocorticoid metabolism [49]. Another study also reported that chlorogenic acid and hypericin can exert antidepressant effects via the gut microbiota-neuroinflammatory axis [50]. These above studies suggested that hypericin may produce antidepressant effects through multiple potential mechanisms. However, it is unclear whether the expression of key genes and molecular targets in those hypericin related mechanisms are regulated by m6A modifications, which warrants further investigation.

  1. The Conclusion section should not just repeat the study results; it should provide a summary of key study findings (positive and negative) and discuss their implications for research and practice.

Response: Thanks. We added the significance of our study and the subsequent outlook in the conclusion section on line 321-330. The revised parts are listed as follows:

In conclusion, our study has found that hypericin improved depression-like behaviors in UCMS mice, while upregulating METTL3 and WTAP expression in the hippocampus of UCMS mice, and the neurotrophin signaling pathway through m6A modification to exhibit antidepressant effects. The present study suggested that the m6A modifying enzyme METTL3 as well as WTAP may play a key role in the pathogenesis as well as the treatment of depression, which provided some hints for the development of new antidepressant drugs from the epitranscriptomic level. In addition, another important finding is that the antidepressant efficacy of hypericin is closely related to the neurotrophic factor signaling pathway, suggesting that the neurotrophic factor signaling pathway may be a potential mechanism for antidepressants.

  1. A substantial degree of language editing is needed to address errors in spelling ("duloxetine" for "doluxetine", "relieve" for "relive") and grammar.

Response: Thanks. We have re-edited the language in the article and corrected the corresponding spelling and grammatical errors.

Round 2

Reviewer 1 Report

The authors extensively edited and nicely reflected suggestions. Overall, the results are easier to follow and highlight their findings. 

The authors should add references on lines 115-118. Not all the major writers or erasers were implicated in the pathology of depression and some studies suggest different results. The authors could clarify the previous finding compared to their results. 

Author Response

Comment: The authors should add references on lines 115-118. Not all the major writers or erasers were implicated in the pathology of depression and some studies suggest different results. The authors could clarify the previous finding compared to their results.

Response: We appreciated this comment. As suggested by reviewer, the reference have been added into the results parts (on line 119-120). We searched for the literatures on writers or erasers in the study of depression pathology and compared the results in previous studies and our results on line 117-130 and 137-142. The revised paragraphs are listed as follows:

Considering that METTL3, METTL14, and WTAP are core components of the methyltransferase complex, and FTO and ALKBH5 are major demethylases [11,13], it has been reported that METTL3, FTO and ALKBH5 were implicated in the pathology of depression [15,18,23]. Therefore, those five genes were selected as potential targets. Meanwhile, there have not been consistent results of the changes in these potential genes (METTL3 and FTO) and more studies need to be validated. For METTL3, Xu et al. (2022) found that the METTL3-specific deletion in the mouse hippocampus can induce the phenotype of depression-like behaviors and spatial memory reduction [18]. In contrast, another study on METTL3 reported to significant upregulation in the hippocampus of UCMS rats and contributed to their cognitive deficits (2022) [24]. For the inconsistent findings of FTO, downregulation expression of FTO was found in the hippocampus of depressed patients and mice exhibiting depressive-like behaviors [15]. However, in another study, the FTO deficiency mice model may reduce anxiety and depressive-like behavior via changes in the gut microbiota [25].

In the present study, we replicated the downregulation of METTL3 in the hippocampus tissue in UCMS mice, which was consistent with the finding that METTL3-specific deletion in the mouse hippocampus can induce depressive-like behaviors [18]. Interestingly, we also discovered that the expression of WTAP decreased significantly in the mice with depressive-like behavior, which have not shown in the previous studies.

Reviewer 2 Report

The authors have extensively revised the manuscript and the text has been greatly improved as a result. It is now easy to follow even for someone who, like this reviewer, has no experience with the molecular biological techniques described. 

The English language has also been carefully considered, but unfortunately some warped wording and half sentences are still present. It might be advisable to have the text edited by the service offered by MDPI. 

The large number of abbreviations may deter the reader. Reference would suggest that the supplementary files begin with a list of abbreviations and refer to them in the introduction to the article. 

Figure 3C could possibly still be explained in a single sentence (in line 139 and in the legend of Figure 3 the letter 'C' is mentioned twice.

Supplementary Tables 1, 2 and 3 were no longer found by referent in the supplementary file. 

For the connections between the hippocampus and habenula, it is best to also refer to Loonen and Ivanova, 2022 (doi: 10.1017/neu.2022.15).

The reference to the supplementary files for more details of the methods used is best placed at the beginning of the methods and materials section. 

Please include subheadings in the supplementary file for the tables and figures exactly as is currently done for the supplementary methods and materials section.

The relationship between the antidepressant effects of ketamine and the expression of BDNF, as well as the strong effects of ketamine in the habenula, makes referent very curious about the results, if instead of hippocampal, habenular tissue was to be examined. The relationship with neuroinflammation also makes genes related to immune activation interesting. These may be topics for future research.

Author Response

Comment 1: The English language has also been carefully considered, but unfortunately some warped wording and half sentences are still present. It might be advisable to have the text edited by the service offered by MDPI.

Response: Thanks for your suggestion, we edited the language of this manuscript via MDPI service, and put the certificate of the MDPI service into the supplementary files.

Comment 2: The large number of abbreviations may deter the reader. Reference would suggest that the supplementary files begin with a list of abbreviations and refer to them in the introduction to the article.

Response: Thanks for your suggestion, we added a list of abbreviations in the supplementary files begin and referred to them in the introduction to the article.

Comment 3: Figure 3C could possibly still be explained in a single sentence (in line 139 and in the legend of Figure 3 the letter 'C' is mentioned twice.

Response: Thanks, we added a single sentence to explain Figure 3C on line 157-158. We corrected the legend of Figure 3C. The revised paragraphs are listed as follows:

The majority of m6A peaks were preferentially located in the 3’UTR and CDS region (Figure 3C).

Comment 4: Supplementary Tables 1, 2 and 3 were no longer found by referent in the supplementary file.

Response: Thanks, we added supplementary Tables 1, 2 and 3 in the supplementary files.

Comment 5: For the connections between the hippocampus and habenula, it is best to also refer to Loonen and Ivanova, 2022 (doi: 10.1017/neu.2022.15).

Response: Thanks, we added citations to the relevant literature on line 276 and 279. The revised paragraphs are listed as follows:

It has been noted that the hippocampus connected to medial habenula (MHb) via the hypothalamus and septal nuclei [32]. This connection of hippocampal complex and the lateral habenula (LHb) may involve into the projections of GABAergic from the lateral septum [33-35].

Comment 6: The reference to the supplementary files for more details of the methods used is best placed at the beginning of the methods and materials section.

Response: Thanks, we have placed the supplementary files for more details of the methods at the beginning of the methods and materials section on line 369-370. The revised paragraphs are listed as follows:

All the experimental details can be found in the Supplementary Materials and Methods.

Comment 7: Please include subheadings in the supplementary file for the tables and figures exactly as is currently done for the supplementary methods and materials section.

Response: Thanks, we added subheadings in the supplementary file for the tables and figures.

Comment 8: The relationship between the antidepressant effects of ketamine and the expression of BDNF, as well as the strong effects of ketamine in the habenula, makes referent very curious about the results, if instead of hippocampal, habenular tissue was to be examined. The relationship with neuroinflammation also makes genes related to immune activation interesting. These may be topics for future research.

Response: Thanks, we added a discussion of the possible relationship between the antidepressant effects of ketamine, the expression of BDNF and habenular on line 317-324. And also discussed the possible links between neuroinflammation, immune activation and the pathology of depression on line 344-352. The revised paragraphs are listed as follows:

Meanwhile, as the center of reward and aversion, the habenula tissue has been implicated into the pathophysiology of major depression and antidepressant effects [49]. In rat and mouse models of depression, it has been reported that antidepressant actions of ketamine mainly mediate the blockade of bursting activity in the lateral habenula neurons and relieve behavioral despair and anhedonia [50]. Additionally, Carolin Hoyer et al. (2012) found that deep brain stimulation treatment of the lateral habenula increased BDNF serum levels and exhibited an antidepressant effect [51]. This suggested that NTRK2 and BDNF may play a key role in habenula and could be associated with the antidepressant efficacy of ketamine.

Additionally, neuroinflammation can sustain the activation of the brain immune cell microglia and recruit other immune cells into the brain, a process which is typically associated with depression [59,60]. Neuroinflammation and immunological responses can be modulated by the STING gene and its pathways in the central nervous system. Administration of STING agonist can ameliorate stress-driven depression-like behaviors through the activation of microglial phagocytosis and suppression of neuroinflammatory cytokines [61]. The use of immunomodulatory ibrutinib also can alleviate neuroinflammation and synaptic defects and have antidepressant effect in the LPS-induced depressive-like behavior mice [59].

Reviewer 3 Report

The revisions made by the authors are satisfactory in my opinion. I have no further major changes or corrections to suggest.

Author Response

Thanks again for your suggestions.